# Longitudinal relationships between habitual physical activity and pain tolerance in the general population

Anders Pedersen Årnes[1]*‡*, Christopher Sievert Nielsen[2,3]◉, Audun Stubhaug[3,4]◉, Mats Kirkeby Fjeld[2], Aslak Johansen[1], Bente Morseth[5]◉, Bjørn Heine Strand[2,6‡], Tom Wilsgaard[7‡], Ólöf Anna Steingrímsdóttir[2]◉

**1** Department of Pain, University Hospital of North Norway, Tromsø, Norway, **2** Division of Mental and Physical Health, Norwegian Institute of Public Health, Oslo, Norway, **3** Division of Emergencies and Critical Care, Department of Pain Management and Research, Oslo University Hospital, Oslo, Norway, **4** Institute of Clinical Medicine, University of Oslo, Oslo, Norway, **5** School of Sport Sciences, UiT The Arctic University of Norway, Tromsø, Norway, **6** The Norwegian National Centre for Ageing and Health, Vestfold Hospital Trust Norway, Tønsberg, Norway, **7** Department of Community Medicine, UiT The Arctic University of Norway, Tromsø, Norway

◉ These authors contributed equally to this work.
‡ APÅ, BHS and TW also contributed equally to this work.
* anders.arnes@uit.no

**Citation:** Årnes AP, Nielsen CS, Stubhaug A, Fjeld MK, Johansen A, Morseth B, et al. (2023) Longitudinal relationships between habitual physical activity and pain tolerance in the general population. PLoS ONE 18(5): e0285041. https://doi.org/10.1371/journal.pone.0285041

**Data Availability Statement:** The data for this study cannot be shared publicly because the current study is based on data owned by a third party (The Tromsø Study, Department of

## Abstract

Physical activity (PA) might influence the risk or progression of chronic pain through pain tolerance. Hence, we aimed to assess whether habitual leisure-time PA level and PA change affects pain tolerance longitudinally in the population. Our sample (n = 10,732; 51% women) was gathered from the sixth (Tromsø6, 2007–08) and seventh (Tromsø7, 2015–16) waves of the prospective population-based Tromsø Study, Norway. Level of leisure-time PA (sedentary, light, moderate, or vigorous) was derived from questionnaires; experimental pain tolerance was measured by the cold-pressor test (CPT). We used ordinary, and multiple-adjusted mixed, Tobit regression to assess 1) the effect of longitudinal PA change on CPT tolerance at follow-up, and 2) whether a change in pain tolerance over time varied with level of LTPA. We found that participants with high consistent PA levels over the two surveys (Tromsø6 and Tromsø7) had significantly higher tolerance than those staying sedentary (20.4 s. (95% CI: 13.7, 27.1)). Repeated measurements show that light (6.7 s. (CI 3.4, 10.0)), moderate (CI 14.1 s. (9.9, 18.3)), and vigorous (16.3 s. (CI 6.0, 26.5)) PA groups had higher pain tolerance than sedentary, with non-significant interaction showed slightly falling effects of PA over time. In conclusion, being physically active at either of two time points measured 7–8 years apart was associated with higher pain tolerance compared to being sedentary at both time-points. Pain tolerance increased with higher total activity levels, and more for those who increased their activity level during follow-up. This indicates that not only total PA amount matters but also the direction of change. PA did not significantly moderate pain tolerance change over time, though estimates suggested a slightly falling effect possibly due to ageing. These results support increased PA levels as a possible non-pharmacological pathway towards reducing or preventing chronic pain.

Community Medicine, UiT The Arctic University of Norway). Ethical and legal restrictions prevent data from being made publicly available. These revolve around the protection of potentially sensitive data that cannot be shared publicly without being in risk of breaching data protection laws. Bona fide researchers can apply for data from the Tromsø Study. Guidelines on how to access the data are available at the website https://uit.no/research/tromsostudy. All inquiries about the Tromsø Study should be sent by e-mail to tromsous@uit.no. Similar reasoning has been given for previous publications of Tromsø Study-based studies in PLOS ONE.

**Funding:** APÅ was funded by a grant from the Northern Norway Regional Health Authority (grant number HNF1352-17). www.helse-nord.no. The funders had no role in study design, data collection and analysis, decision to publish, or preparation of the manuscript.

**Competing interests:** The authors have declared that no competing interests exist.

## Introduction

Physical activity (PA) is a commonly recommended non-pharmacological intervention for preventing and treating a range of chronic pain conditions [1–7]. Concurrently, the prevalence of chronic pain and musculoskeletal complaints is seen to decrease with higher levels of PA in cohort studies [8–11]. There is some evidence regarding a pain-inhibitory response immediately following an acute bout of exercise. This phenomenon is referred to as exercise-induced hypoalgesia (EIH), and was reviewed with regards to exercise protocols, possible mechanisms, and behaviour in sub-populations by Rice et al. in 2019 [12]. Although evidence is sparse, results from experimental studies indicate that the presence of chronic pain can lower the efficacy of EIH [12, 13]; i.e. reducing potential effects of exercise on pain sensitivity. As in acute exercise, higher levels of *habitual* PA are also associated with lower sensitivity to experimental pain [14–17]. Some studies have suggested that individual sensitivity to some quantitative sensory tests of pain has predictive value for subsequent development and progression of chronic pain, often post-operatively [18–22], but the evidence is conflicted and frequently suffers methodological challenges regarding quality of studies and choices of exposures and outcomes. In summary, the sparse literature in this field indicates that a reduction in pain sensitivity might be a possible mechanism through which higher habitual PA levels might modify the risk, or progression, of chronic pain.

Previous studies of PA and pain sensitivity commonly employ small, homogenous samples of young, healthy, or single-sex subjects. In a review by Tesarz et al. including 15 studies of between 6 to 67 participants, athletes had significantly higher pain tolerance than normally active controls, but data were less uniform regarding pain detection thresholds [16]. Several of the studies were single-sex samples and most were on students <30 years of age. Two later studies (n = 53 and n = 36) further supported such an association to pain tolerance in athletes in particular [23, 24]. However, little basic research exists to describe the relationship between habitual PA and pain tolerance in the general population. Our recent cross-sectional study on approximately 19,000 participants was the first study with a sample size of this magnitude to find that higher population-based levels of habitual PA were similarly associated with higher cold-pressor pain tolerance in the general population as that seen in smaller observational and experimental studies [25]. However, causal direction cannot be ascertained by cross-sectional studies. Interestingly, two experimental studies on 24 and 20 healthy participants found increases in pain tolerance following a 6-week moderate to high exercise intervention [14, 26], indicating an effect on pain tolerance by leisure-time types of PA. However, these were of low power and unable to investigate conditional effects for sex and clinical pain. As large studies on PA interventions are lacking, a population-based approach to assessing whether a population change of PA is related to subsequent pain tolerance could provide important basic knowledge.

Furthermore, it would be relevant to examine whether PA influences any potential change in pain tolerance when measured repeatedly in the same individuals over time, and also how these longitudinal relationships are affected by moderating factors such as sex and clinical pain.

Using population data from the Tromsø Study, our current objectives were thus 1) to assess the relationship between longitudinal habitual PA change and subsequent pain tolerance, and 2) to estimate the longitudinal relationship between habitual PA and pain tolerance in repeated measurements of individuals and assessing whether PA moderated any change in tolerance over time. We also assessed whether these relationships changed over sex or chronic pain status.

## Materials and methods

### Ethics

This study was approved by the Regional Ethics Committee of North-Norway (case number REK North, 2016/1794). Written informed consent was acquired for all participants.

### Study population and sample

The present study used data from the sixth and seventh surveys of the Tromsø Study: Tromsø6 (baseline, years 2007–08) and Tromsø7 (follow-up, years 2015–16). The Tromsø Study is a prospective population-based health study conducted in the municipality of Tromsø, Northern Norway. It has gathered population-wide data on PA and experimental pain tolerance in two surveys separated by 7–8 years. This includes data on potentially confounding or moderating factors, including sex, chronic pain, and socio-demographic covariates, and is the largest source of repeated measurements of quantitative sensory test data in the world. Such data can be used to assess relationships with temporal ordering of events. Total birth cohorts and random samples of the local populace have been invited to participate through mailed invitations. No payment is offered for participation. The study collects data through questionnaires, biological samples, and clinical examinations. Further information about recruitment and participation proportions for the entire study has been given elsewhere [27–29].

In Tromsø6, 66% of invitees participated (n = 12,984; mean age 57.5 years; 53% women), while participation proportion for Tromsø7 was 65% (n = 21,083; mean age 57.3 years; 53% women). Of all participants in Tromsø6, 11,284 were especially invited to a follow-up visit in Tromsø7, which 79% attended (n = 8,906; mean Tromsø6 age 55.8 years; 54% women). Both Tromsø6 and Tromsø7 included questionnaires on physical activity and quantitative sensory testing of pain using several types of modalities. The current study sample included individuals participating in both Tromsø6 and Tromsø7 who had information on PA and cold-pressor test (CPT) tolerance at baseline and follow-up (Fig 1; n = 10,732).

### Measurements and variables

**Leisure-time physical activity.** Participants self-reported LTPA level in both surveys using a modified version of the four-level "Saltin and Grimby LTPA Physical Activity Level Scale" (SGPALS [30, 31]). SGPALS asks participants to recall the past 12-month-average level of LTPA specifying four mutually exclusive categories: "Reading, watching TV, or other sedentary activity"; "walking, cycling, or other forms of exercise at least four hours a week (with examples)"; "participation in recreational sports, heavy gardening, etc. at least four hours a week"; or "participation in hard training or sports competitions, regularly several times a week". Categories correspond to sedentary, light, moderate, or vigorous LTPA.

**The cold-pressor test.** CPT pain tolerance was measured on-site at baseline and follow-up as maximum tolerance time during the CPT. Participants placed their dominant (Tromsø6) or non-dominant (Tromsø7) hand and wrist in a 13-litres Plexiglass vat containing water maintained at 3.0˚C by a cooling circulator (Julabo FP40HE, Julabo Labortechnik GmbH). The difference in test-methodology was due to the addition of an electronic VAS rating mechanism in Tromsø7 which had to be operated using the dominant hand.

During testing, participants were asked to keep their hand open and relaxed with the hand and wrist submerged in the water for as long as possible, up to a maximum tolerance time of 106 seconds for Tromsø6 and 120 seconds for Tromsø7. Participants were informed of the possibility to abort the test at any time during testing.

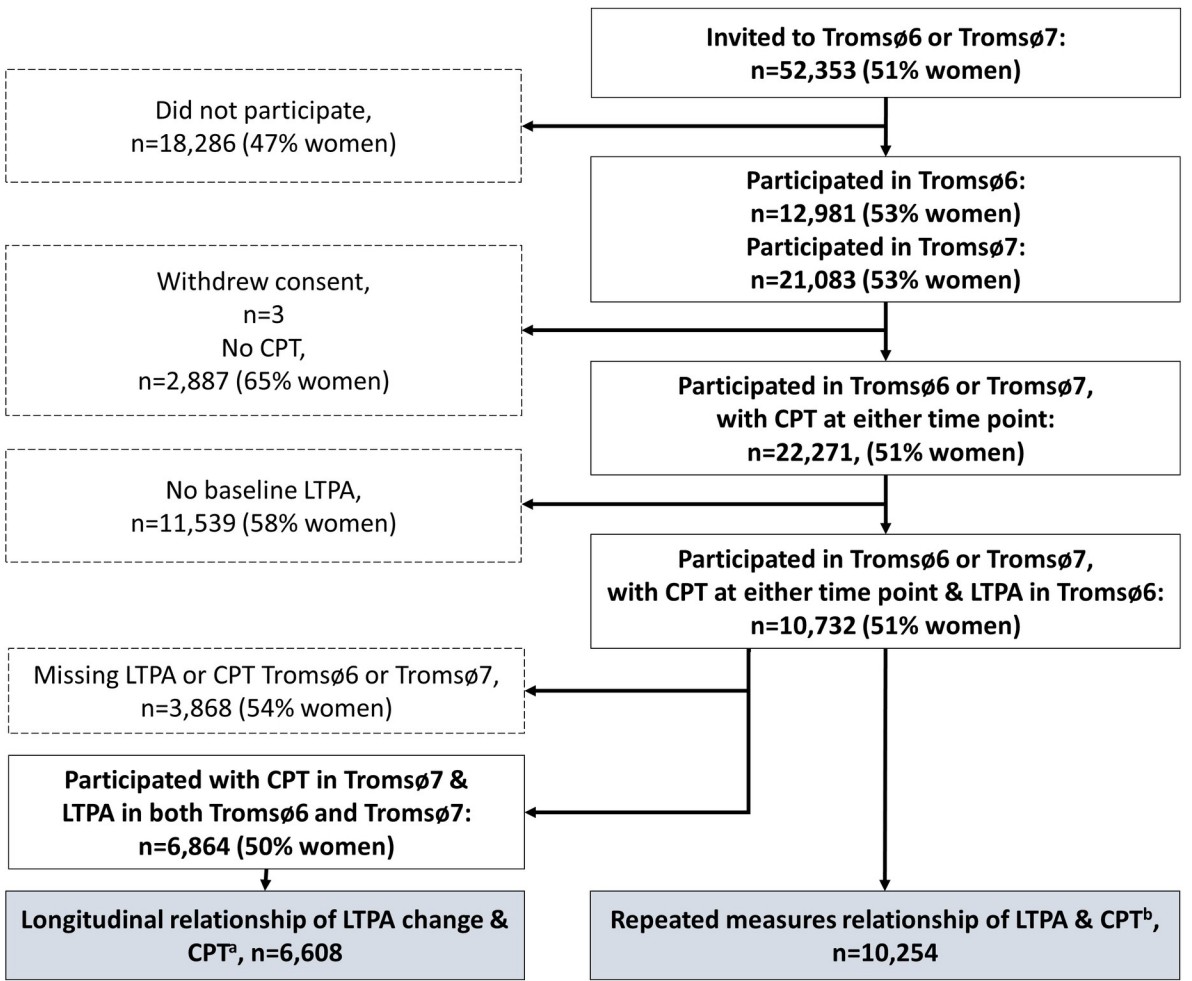

**Fig 1. Flow of study participants.** [a] Linear Tobit regression; n missing covariates = 256. [b] Mixed model Tobit regression; missing on covariates = 478. LTPA = leisure-time physical activity; CPT = cold-pressor test.

Participants were excluded from CPT in Tromsø6 or Tromsø7 according to the following criteria: unwilling to participate; cognitive or language problems making them unable to comprehend and follow instructions; Reynaud's syndrome, cold allergy or other conditions that in participants' experience affects their response to cold; bilateral loss of sensitivity in the hand; breached skin on both hands (e.g. caused by eczema, open sores).

We recoded maximum tolerance times for CPT in Tromsø7 to 106 s. post hoc to make the censoring time identical for both surveys. This recoded the right-censored values of 120 s. to 106 s. for 2,499 participants of Tromsø7. Of these, 142 participants had CPT values ranging between 107 and 119 sec.

**Baseline covariates.**  Covariates included self-reported level of education (primary or secondary school up to 10 years, technical/vocational/high school up to three years, college/university less than four years, college or university for four years or more); daily smoking (present, previous, or never); alcohol consumption (never, monthly or less frequently, 2–4 times a month, 2–3 times a week, 4 or more times a week); and self-reported health (very bad, bad, neither good nor bad, good, excellent). We also included occupational PA as a covariate as reported by participants on the Saltin and Grimby occupational PA questionnaire: "If you

have paid or unpaid work, which statement describes your work best?". Participants could choose among "Mostly sedentary", "Work that requires a lot of walking", "Work that requires a lot of walking and lifting", and "Heavy manual labour". Participants who did not respond to this but who elsewhere reported being retired or on disability pensions, unemployment benefits, or sick leave were assigned to the categories "retired" or "disability/sick leave", respectively. We also included chronic pain (constant or recurring pain for three months or longer) as a covariate to be able to assess its importance as a possible effect moderator.

These covariates were defined as potential confounders rather than colliders or mediators, based on their previously known or suspected association to physical activity and/or pain sensitivity [32–38]. On the other hand, research regarding occupational PA and pain tolerance is generally lacking; occupational PA was nevertheless expected to be a confounder based on the reported paradoxical relationship between LTPA and occupational pain, and chronic pain and disability [39].

## Statistical analyses

In our primary analysis, we computed an index of LTPA change between baseline and follow-up by computing combinations of LTPA levels across Tromsø6 and Tromsø7. The index was computed as an ordinal variable we assessed the relationship between this index of LTPA change from baseline to follow-up and CPT tolerance at follow-up, using ordinary Tobit regression with right-censored values. We used Tobit regression because the CPT data contained a substantial number of right-censored values (maximum test-time = 106 s.). Such data will bias ordinary linear regression-based estimates of effect. Tobit-class regression models account for the expected distribution of values for the unobserved (here; the right-censored) outcome distribution. Regression parameters can therefore be interpreted as estimates for the true underlying (unbiased) effect on the latent but censored dependent variable, i.e. the expected distribution of the outcome had CPT not been stopped at 106 seconds.

To assess whether a change in pain tolerance over time varied with level of LTPA, our secondary analysis used mixed Tobit regression. Here we estimated the association at both survey occasions, adjusting for survey occasion [40, 41]. Adding a cross-product of LTPA×survey occasion allowed using interaction analysis to assess whether LTPA moderated change of pain tolerance over time. We also added a random intercept for individual subjects to adjust for multiple observations of the same individual due to the repeated measurements of two surveys. In this analysis, we also included participants with only one outcome measurement, as the mixed model used in the secondary analysis makes use of participants with incomplete data to improve the accuracy of estimates. Comparing the model with and without the random intercept for subjects using likelihood-ratio test, we found a significantly better fit (p<0.05) for the random effects model. To evaluate the estimation of the random effects model, we examined the accuracy of the quadrature calculation by doubling the default number of integration points used (14 vs. 7), finding negligible differences in estimates. This suggests high accuracy and thus adequately estimated random effects.

As a sensitivity analysis, we specified an identical model using an ordinary linear mixed model with random intercept to observe the impact on effect estimates of using censored values as they were.

The Tobit model is more vulnerable to assumptions of normality than ordinary least squares regression. We used R [42] package tobitdiag to estimate normal distribution Martingale-type residuals which we plotted and inspected for potential deviations, as suggested by Barros et al. [43, 44]. Results showed some deviation from normality in residuals; we discuss the implications of this under strengths and limitations.

Interactions for LTPA and survey, sex, and chronic pain were assessed by adding cross-products of these variables to separate models and testing their model contribution with likelihood-ratio tests. We also assessed the statistical significance of coefficients of each interaction group.

In both primary and secondary analyses, we adjusted for sex, baseline age, education level, alcohol frequency consumption, self-reported health status, daily smoker status, occupational PA level, and chronic pain to account for their possible confounding effect.

Effect sizes were reported with 95% confidence intervals (CI); significance level was set at 5%. Data analyses were performed using Stata 15 and Stata 16.1 (StataCorp, College Station, TX, USA), and R (R Foundation for Statistical Computing, Vienna, Austria; 42).)

### Missing data

Causes of missing CPT data included program or technician error, as well as 1,831 participants in Tromsø6 who were not tested due to capacity limitations. Whenever this occurred, staff were told to prioritize participants below 60 years of age as that was the age-group under-sampled in the study (Stabell et al., 2013). Individuals not seen at the testing station were regarded as not having participated in CPT.

Of the 6,864 who participated in CPT in Tromsø7 and had two measurements of LTPA, 256 were lost to primary analysis due to missing information on one or more covariates (S1 Table).

Of the 10,752 with baseline LTPA, and CPT in either Tromsø6 or Tromsø7, 478 were lost to analysis due to missing information on one or more covariates (S2 Table).

## Results

The 6,864 participants that reported LTPA in both Tromsø6 and Tromsø7 as well as CPT tolerance in Tromsø7 (50% women; mean age 54.2 (SD 10.8)) were included in primary analyses of LTPA change on subsequent CPT. Furthermore, the 10,732 that participants reported LTPA in Tromsø6 and completed CPT in Tromsø6 and/or Tromsø7 (51% women; mean age 55.8 (SD 11.8)) were included in the overall longitudinal analyses (Fig 1). There was some difference in covariate distributions between men and women (Table 1). Men had a higher age and CPT mean, higher proportion censored in CPT, and proportion engaging in MVPA. Women had the highest proportions of light LTPA, longest education, most chronic pain sufferers, and current retirees. Sample mean CPT outcome over levels of LTPA, sex, and survey occasion is shown in Table 2. There was a general decline in tolerance times across surveys. In both surveys, CPT tolerance was somewhat higher for men vs. women, and higher for higher levels of LTPA.

### LTPA and pain tolerance

In the primary analysis, when using longitudinal LTPA change as exposure and CPT tolerance at follow-up as outcome, we found a statistically significant, positive association for those who remained active over time as compared to those who remained sedentary (Table 3; Fig 2). Effect sizes show increased CPT tolerance primarily for those with the highest total amount of PA; secondly more frequently for those with high vs. low PA level at follow-up; and thirdly to a limited extent for those with a positive vs. a negative change in PA over time. Despite these tendencies in effect estimates, no combination containing sedentary LTPA at any time point was significantly different from those who were sedentary in both surveys. Groups containing combinations of light and moderate-to-vigorous LTPA were statistically similar to each other, with 8–12 s. higher CPT tolerance than those who were sedentary in both surveys. Those

**Table 1. Baseline characteristics of study samples over main analyses models; mixed model by gender.** The Tromsø Study 2007–2016.

| Baseline Characteristic: | Total sample | | | PA-change model |
|---|---|---|---|---|
| | Total sample n = 10,732 | Women n = 5,505 (51%) | Men n = 5,227 | Total sample n = 6,864 |
| Age, mean (SD) | 55.8 (11.8) | 55.3 (12.0) | 56.2 (11.7) | 54.2 (10.8) |
| CPT, mean (SD) | 88.4 (28.3) | 83.4 (30.8) | 93.6 (24.3) | 91.1 (26.5) |
| Censored CPT[a], n; % | 6,718 (62.6) | 3,005 (54.6) | 3,713 (71.0) | 4,369 (63.7) |
| LTPA, n; % | 10,732 (100) | 5,505 (51) | 5,227 (49) | 6,864 (100) |
| *Sedentary* | 19.3 | 18.3 | 20.3 | 17.2 |
| *Light* | 60.2 | 67.7 | 52.3 | 60.3 |
| *Moderate* | 18.8 | 13.1 | 24.8 | 20.4 |
| *Vigorous* | 1.7 | 0.9 | 2.6 | 2.1 |
| Education level, n; % | 10,648 (99.2) | 5,467 (99.3) | 5,181 (99.1) | 6,826 (99.5) |
| *Primary/secondary school* | 24.6 | 26.6 | 22.5 | 20.8 |
| *Technical/vocational/high school* | 34.5 | 33.1 | 35.9 | 35.0 |
| *College less than 4 years* | 18.8 | 16.7 | 21.0 | 20.4 |
| *College 4 years or more* | 22.1 | 23.6 | 20.6 | 23.8 |
| Alcohol consumption, n; % | 10,662 (99.4) | 5,459 (99.2) | 5,203 (99.5) | 6,836 (99.6) |
| *Never* | 8.7 | 10.9 | 6.4 | 6.4 |
| *Monthly or less frequently* | 27.7 | 30.3 | 24.7 | 25.3 |
| *2–4 times a month* | 40.4 | 37.6 | 43.2 | 42.8 |
| *2–3 times a week* | 18.1 | 16.6 | 19.7 | 19.9 |
| *4 or more times a week* | 5.3 | 4.6 | 6.0 | 5.6 |
| Self-reported health, n; % | 10,662 (99.4) | 5,464 (99.3) | 5,198 (99.5) | 6,815 (99.3) |
| *Bad or very bad* | 4.4 | 4.8 | 3.9 | 3.3 |
| *Neither or* | 30.8 | 26.3 | 26.5 | 22.2 |
| *Good* | 83.9 | 51.6 | 54.6 | 55.9 |
| *Excellent* | 16.1 | 17.3 | 14.9 | 18.6 |
| Daily smoker, n; % | 10,615 (98.9) | 5,432 (98.7) | 5,183 (99.2) | 6,810 (99.2) |
| *Yes, now* | 20.2 | 21.7 | 18.6 | 17.5 |
| *Yes, previously* | 42.5 | 39.0 | 46.1 | 43.1 |
| *Never* | 37.3 | 39.3 | 35.2 | 39.4 |
| Chronic pain, n; % | 10,721 (99.9) | 5,499 (99.9) | 5,222 (99.9) | 6,858 (99.9) |
| Yes | 31.7 | 37.1 | 26.0 | 29.9 |
| Occupational PA, n; % | 10,585 (98.6) | 5,430 (98.6) | 5,155 (98.6) | 6,777 (98.7) |
| Sedentary | 39.1 | 35.6 | 42.8 | 43.5 |
| Light | 18.3 | 20.3 | 16.2 | 19.5 |
| Moderate | 13.1 | 13.2 | 12.9 | 13.8 |
| Heavy | 2.3 | 0.8 | 4.0 | 2.3 |
| Retired | 26.4 | 29.3 | 23.4 | 20.2 |
| Disability/sick leave/unemployed | 0.7 | 0.9 | 0.7 | 0.7 |

[a] Censored: Cold-pressor test tolerance = 106 s.

PA = physical activity; SD = standard deviation; CPT = cold-pressor test; LTPA = leisure-time physical activity.

maintaining MVPA across surveys had the highest tolerance time, enduring an estimated 20.4 s. longer than the consistently sedentary reference group.

Weak and overall non-significant interactions were found for sex (Table 3). There was no interaction with chronic pain (results not shown).

In the secondary analyses of CPT change over time, CPT in Tromsø6 and Tromsø7 varied according to level of baseline LTPA (Fig 3). CPT tolerance declined by an estimated average of

**Table 2. Participant mean CPT endurance time (seconds) at both occasions over baseline physical activity levels and sex[a].** The Tromsø Study 2007–2016.

| | n (%) | Leisure-time physical activity | | | |
|---|---|---|---|---|---|
| | | *Sedentary* | *Light* | *Moderate* | *Vigorous* |
| Tromsø 6 | 9,773 | 84.9 (30.2) | 87.7 (28.7) | 93.6 (24.3) | 96.1 (21.2) |
| *Women* | 4,956 (50.7) | 80.0 (31.9) | 83.6 (30.7) | 86.5 (29.5) | 90.2 (25.2) |
| *Men* | 4,817 (49.3) | 89.3 (27.9) | 93.2 (24.5) | 97.5 (20.0) | 98.3 (19.0) |
| Tromsø 7 | 7,136 | 56.6 (37.2) | 60.7 (37.7) | 68.0 (37.2) | 69.0 (37.9) |
| *Women* | 3,605 (50.5) | 52.8 (37.2) | 56.3 (37.6) | 61.6 (38.3) | 60.5 (37.3) |
| *Men* | 3,531 (49.5) | 60.0 (37.0) | 66.7 (37.0) | 71.4 (36.1) | 72.3 (37.9) |

[a]Values are mean CPT tolerance times in seconds with standard deviations in parentheses

CPT = cold-pressor test.

-54.7 seconds from Tromsø6 to Tromsø7 (at means of covariates). For those who were sedentary, this was estimated to be a decline from 122.5 seconds on average in Tromsø6, to 67.8 seconds in Tromsø7.

Overall tolerance time was significantly and positively associated with higher levels of baseline LTPA (Table 4). CPT tolerance was 7%, 14%, and 16% higher respectively for light, moderate, and vigorous habitual LTPA across the two surveys, compared to the sedentary group. The most active participants endured for an estimated average of 16.3 s. (95% CI 6.0, 26.5) longer compared to those who reported being sedentary. There was no statistically significant interaction between LTPA and survey occasion, indicating that the change in pain tolerance over time did not differ according to level of baseline LTPA (Table 4). However, the

**Table 3. Regression coefficients with 95% confidence limits for the association between leisure-time physical activity change over time and cold-pressor tolerance time (seconds) overall and by sex.** The Tromsø study 2007–2016.

| LTPA change index[a] | n = 6,608 | Overall | Women | Men |
|---|---|---|---|---|
| Reference group CPT tolerance[c] | 477 | 64.6 (59.4, 69.9) | 63.2 (55.7, 70.8) | 66.7 (59.6, 73.9) |
| *Sedentary-Sedentary* | 477 | 0 (reference) | 0 (reference) | 0 (reference) |
| *Light-Sedentary* | 366 | 4.4 (-3.5, 12.3) | 3.2 (-8.0, 14.5) | 5.1 (-5.8, 16.0) |
| *Sedentary-Light* | 532 | 6.1 (-1.0, 13.2) | 1.4 (-8.8, 11.5) | **10.1 (0.2, 20.1)** |
| *Sedentary-MVPA* | 114 | 9.0 (-2.9, 20.8) | 0.1 (-19.4, 19.6) | **15.9 (0.9, 30.1)** |
| *MVPA-Light* | 545 | **10.8 (3.6, 18.1)** | 3.8 (-7.2, 14.7) | **16.8 (7.2, 26.4)** |
| *Light-Light* | 2,868 | **11.3 (5.7, 17.0)** | 4.9 (-3.2, 12.9) | **17.1 (9.2, 25.1)** |
| *Light-MVPA* | 759 | **11.9 (5.2, 18.7)** | 0.7 (-8.8, 10.3) | **22.7 (13.4, 32.0)** |
| *MVPA-Sedentary* | 52 | 15.6 (-1.3, 32.5) | 14.3 (-15.2, 43.8) | 18.8 (-2.0, 39.5) |
| *MVPA-MVPA* | 895 | **20.4 (13.7, 27.1)** | **13.1 (2.8, 23.5)** | **26.2 (17.5, 34.9)** |
| *p*-value for equality[d] | | **<0.001** | | |
| *p*-value for equality[e] men vs. women | | | | 0.0732 |

[a] Linear Tobit regression with upper limit (censoring) = 106 s.

[b] Significant interaction levels in **bold**.

[c] Model-predicted mean of CPT tolerance for reference group at means of covariates.

[d] Global Wald test of equality between all coefficients.

[e] Test of interaction between LTPA and sex using likelihood ratio test.

Models adjusted for baseline sex, age, education, alcohol consumption frequency, smoking status, self-reported health, occupational physical activity, chronic pain.

Significant results in **bold**.

Abbreviations: LTPA = leisure-time physical activity; CPT = cold-pressor test; CI = confidence interval; MVPA = moderate-to-vigorous physical activity.

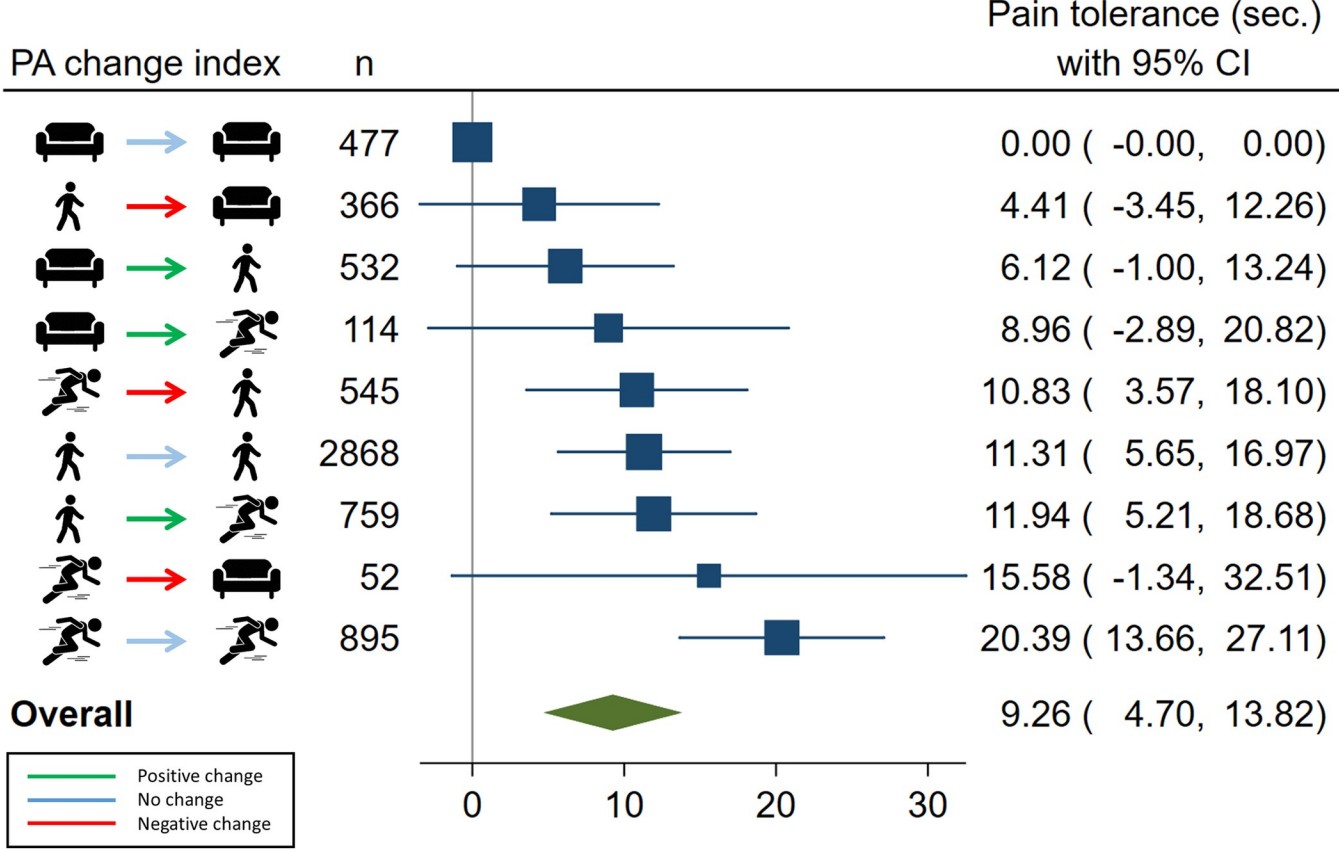

**Fig 2. Relationship between groups of physical activity change from Tromsø6 to Tromsø7 and seconds of cold pain tolerance.** Ordered by effect size. PA = physical activity; CI = confidence interval.

interaction was borderline significant when modelling LTPA as a continuous rather than a categorical variable, and subsequently testing the linear trend and effect estimates suggested a gradually increasing negative trend.

There was no significant interaction with sex, although tolerance appeared to be higher for males, and no interaction with chronic pain (S3 Table).

Using ordinary linear, rather than Tobit, mixed regression appeared to substantially underestimate effect sizes, although results remained statistically significant. E.g.: linear models would underestimate the effect estimate of vigorous LTPA by almost 60% (6.7 vs. 16.3 s.; S4 Table).

## Discussion

In this study, pain tolerance increased with level of PA. Being physically active at either of two time points measured at a 7-8-year interval was associated with higher pain tolerance compared to being sedentary at both time-points. Pain tolerance increased with higher total activity levels, and more for those who increased their activity level at follow-up. Overall, higher LTPA was associated with a significantly higher pain tolerance when measured repeatedly in the same individuals. A general decline in pain tolerance over the two time points was not significantly moderated by the level of LTPA, although the benefit of higher levels of LTPA on pain tolerance seemed to be gradually decreasing over time.

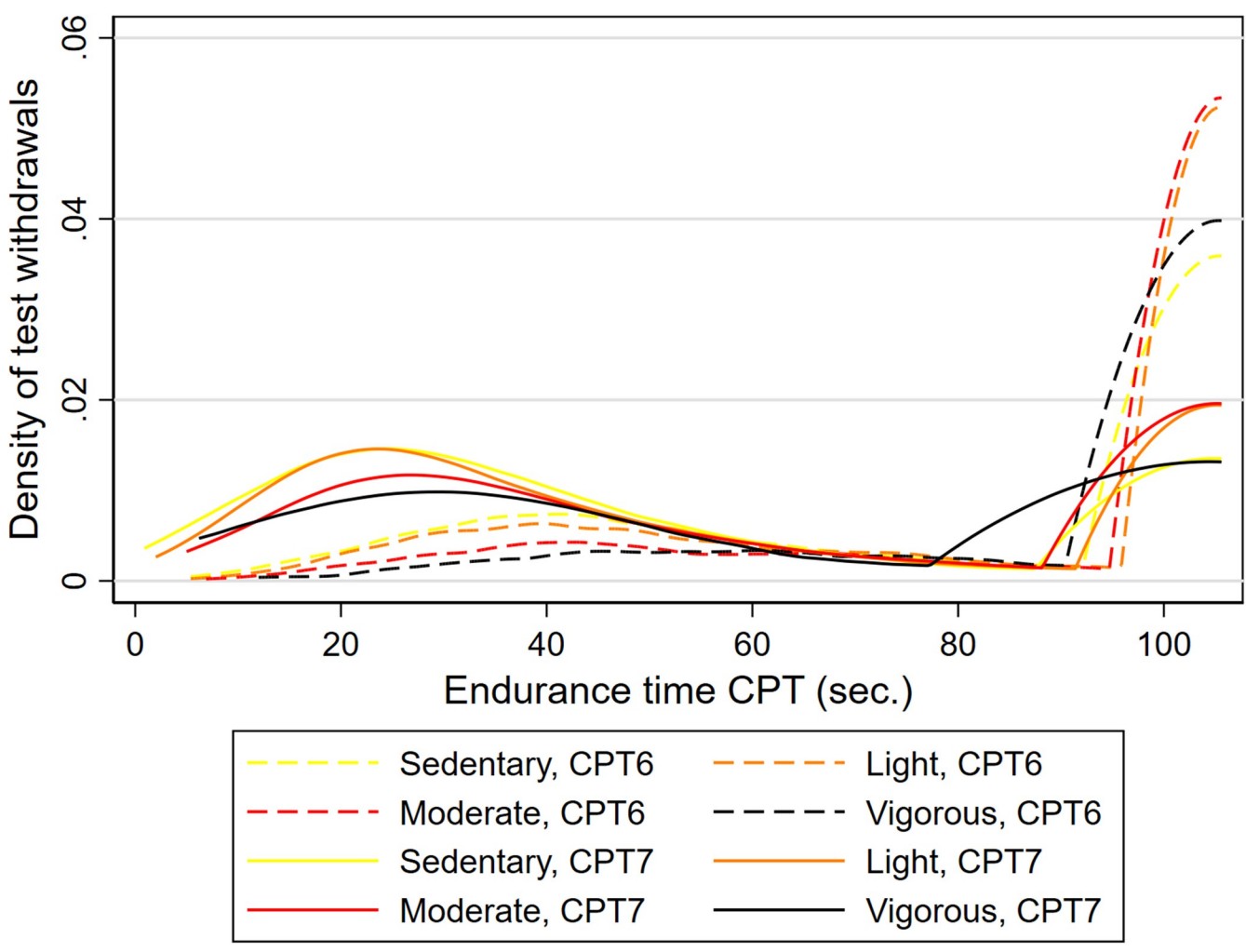

**Fig 3. Withdrawals from the cold-pressor test according to leisure-time physical activity groups.** Tromsø6 and Tromsø7. CPT = cold-pressor test; LTPA = leisure-time physical activity (6 or 7 for respective Tromsø Study survey).

### Physical activity and cold-pressor test tolerance

**Physical activity changes and pain tolerance.** In comparison to the present results, two small clinical studies have indicated that inducing PA change in humans over time may increase pain tolerance: Exposing 24 healthy participants to a high-intensity cycle ergometer program for 6 weeks caused ischemic pain tolerance to increase by 20%, with no increase in the normally active controls [14]. O'Leary et al. corroborated this in 6 weeks for high-intensity interval training only [26]. They theorized that the intensity required could be because the noxious stimulus produced by the metabolic disturbance inherent in high-intensity exercise causes a familiarization and subsequent shift in pain tolerance levels. They further found no evidence of this being linked to an improvement in physical fitness levels. A meta-analysis of 15 randomized controlled trials likewise found adaptations of pain sensitivity thresholds to occur over time in exercise interventions in both healthy individuals and individuals with chronic pain [45].

Mechanisms through which such PA change might influence pain sensitivity in humans are poorly understood. As most studies have investigated acute exercise-induced hypoalgesia

**Table 4. Regression coefficients with 95% confidence intervals for the association between baseline levels of leisure-time physical activity and cold-pressor tolerance time (seconds) without and with time interaction.** The Tromsø Study 2007–2016.

| | n = 10,254 | Model 1[a]: Overall | Model 2[b]: Baseline CPT | Model 2[b]: CPT change |
|---|---|---|---|---|
| Reference group CPT tolerance[c] | 1,962 | 99.4 (96.5, 102.3) | 122.5 (119.1, 125.9) | -54.7 (-58.2, -51.2) |
| Baseline LTPA | | | | |
| *Sedentary* | 1,962 | 0 (reference) | 0 (reference) | 0 (reference) |
| *Light* | 6,178 | **6.7 (3.4, 10.0)** | **6.7 (2.9, 10.5)** | -0.01 (-4.1, 4.1) |
| *Moderate* | 1,933 | **14.1 (9.9, 18.3)** | **16.6 (11.6, 21.6)** | -4.6 (-9.8, 0.6) |
| *Vigorous* | 181 | **16.3 (6.0, 26.5)** | **20.0 (7.3, 32.8)** | -6.6 (-19.5, 6.3) |
| *p* for trend | | **<0.001** | | 0.054 |
| *p* for equality[d] | | | | 0.13 |

[a] Mixed model Tobit regression with upper limit (censoring) = 106 s. for latent distribution of CPT outcome. Models were adjusted for measurement occasion, as well as baseline sex, age, and self-reported occupational PA level, education, alcohol consumption frequency, smoking status, health status, and chronic pain. Significant results in **bold**.

[b] Mixed model with LTPA×survey interaction.

[c] Model-predicted sedentary CPT tolerance at means of covariates.

[d] Test of interaction between LTPA and time using the likelihood ratio test.

Abbreviations: LTPA = leisure-time physical activity; CI = confidence interval; CPT = cold pressor test.

(EIH), the underlying mechanistic hypotheses mostly relate to this phenomenon. These include PA-induced activation of endogenous opioid and endocannabinoid modulation of pain, and genetic, immunological and psychological mechanisms [12]. On the other hand, the 'chronic' effect of habitual PA level on pain sensitivity has garnered less attention, perhaps mainly through animal models of EIH. In a recent review of animal studies, regular physical activity appeared consistently effective at reducing, or preventing, hyperalgesia in neuropathic, and inflammatory and non-inflammatory muscle pain models [46]. Some of these mechanisms observed in animal studies appear to overlap those proposed in humans, like the mediation by mu-opioid receptors of analgesia induced by habitual wheel running in mice [47].

It is important to assess whether these patterns primarily express the pain tolerance required to tolerate physical activity at certain levels, or if a PA change in humans can lead to a change in pain tolerance. Contrary to O'Leary et al. [26], our modelling of PA change and pain tolerance at follow-up primarily found the greatest effect in avoiding a persistently sedentary lifestyle. This resonates with the idea that a sedentary lifestyle has a detrimental impact on health in general [48, 49]. The results further indicate that a change to or away from being sedentary yielded higher effect estimates than remaining sedentary. Also, higher total, as well as consistent, amounts of PA reported over time appeared to be positively associated with pain tolerance compared to remaining sedentary. These effect estimates were dose-response shaped for consistent light PA and moderate-to-vigorous PA in a way similar to that reported in a previous cross-sectional study [25]. Notably, participants changing one PA level over time were not significantly different from those that kept a consistent level. This similarity could be due to sensitivity issues with the questionnaire, a lack of statistical power in the model, or possibly that the change had not yet had the time to impact pain tolerance. Finally, though levels of change did not have unequivocal patterns of association to pain tolerance, increasing PA level appeared to predict stronger associations to pain tolerance than a decrease. The latter was always associated with a smaller effect estimate than maintaining or increasing PA beyond the original level. This might indicate that the direction of change matters in addition to total amount of activity.

In summary, these findings suggest that becoming or remaining active at a level above being sedentary, or making a positive change in activity level, over time is associated with higher pain tolerance as opposed to being sedentary or making a negative change.

**The stability of the relationship over time.**   The secondary analyses of this study aimed at assessing whether pain tolerance changed for the included individuals over time, and whether any such change was moderated by their level of LTPA. This is the first population-based study to estimate the repeated association of LTPA level and pain tolerance, and to assess how a change in pain tolerance over time was moderated by habitual LTPA. The repeated measurements-association between PA and CPT tolerance was similar to results from our recent cross-sectional study using total samples drawn from Tromsø6 and Tromsø7 [25].

The lack of significant interaction between LTPA and time indicates that baseline PA level did not significantly influence the general drop in pain tolerance across the two measurements of individuals over time. However, though this interaction was not significant, the linear trend of moderation, as well as effect estimates, might suggest that the positive association of LTPA and pain tolerance diminishes in size over time, and more so for higher activity groups. This interaction between LTPA and time might have gained significance with higher power in the highest PA groups.

Our study sample consisted of individuals aged 30–87 years at baseline, with approximately eight years separating the two survey occasions. Thus, it is possible that ageing interferes with the association of LTPA and pain tolerance, potentially diminishing a positive effect over time. Whether ageing interferes with the effect of LTPA on pain tolerance, especially in older age groups, is something which should be further explored in future studies. Alternative explanations to this time-effect could be methodological differences between Tromsø6 and Tromsø7 of which effect we are not aware.

**Potential moderators.**   Several studies of both humans and animal models have identified sex as one of the determinants of pain sensitivity or modulation [32, 33, 50, 51]. In our previous cross-sectional study we also found that sex moderated the PA-tolerance relationship [25]. Despite some signs of sex differences in the effect estimates of our PA-change model, no overall significant interaction was seen in our current study.

There is inconsistent evidence regarding EIH in patients with chronic pain, in part due to a lack of high-quality studies [12, 13]. A narrative review suggested no EIH in patients with localized musculoskeletal pain, however only reviewing isometric exercise and sensitivity thresholds [52]. Nevertheless, using both the standard 3-month cut-off for chronic pain as in the present study, and a stricter 'moderate-to-severe chronic pain' definition previously, chronic pain has not influenced the association of habitual PA and pain tolerance in a general population either in cross-sectional designs, longitudinally, or when looking at PA change over time. This suggests that the present epidemiologically defined chronic pain does not significantly interfere with the relationship between PA and pain tolerance in large heterogeneous samples. Naturally, this might look different in more highly selected diagnostic groups or if using different definitions of chronic pain.

## Possible limitations

The observational and temporal nature of these data obscure how the exposure, covariates, and outcome vary prior to baseline, and between baseline and follow-up. As we did not adjust for baseline CPT in our model in order to avoid the bias expressed as Lord's paradox [53], part of the associations observed in our PA-change model might theoretically express some dynamic of pain tolerance during follow-up. However, sensitivity analysis with adjustment (results not shown) found negligible change in associations and only slightly diminished effect estimates.

Exploratory analyses found a significant interaction between CPT tolerance and survey. As our models look at relative group difference rather than absolute tolerance levels, this difference is not likely to impact results.

Whilst self-report tools like the SGPALS may over- or under-report absolute amount of PA undertaken, they have consistently proven to adequately rank respondents according to health outcomes, thus being suitable for group comparisons [31, 54]. Furthermore, the SGPALS aims to capture physical activity over a 12-month period rather than the relatively short time span used by other questionnaires or methodologies. This may give more accurate grouping of participants in longitudinal data. However, the similar effect estimates of several PA change categories might indicate that the SGPALS is inaccurate when measuring amounts of PA change over time; some participants might define themselves as bordering two categories. Their change score might reflect this more than any actual PA change.

Our use of Tobit regression on quantitative sensory test data suggests how high proportions of censored data may bias effect estimates of pain tolerance means. Since we discovered some deviations from normally distributed residuals, borderline p-values have to be interpreted with care. However, most of the current significant results had very low p-values, and high statistical power in analyses further diminishes the risk of miscalculated p-values impacting significance.

## Conclusion

In this study of a general population sample, being physically active across two measurements was associated with higher pain tolerance at follow-up as compared to being sedentary at both time-points. Furthermore, changing PA from lower to higher levels might be associated with a higher pain tolerance than an equally large change going from higher to lower PA. This might indicate that it is not only the total PA amount that matters but also the direction of change. Repeated measurements of this association in the same individuals over two time points found a negative change in pain tolerance over time that was not significantly moderated by LTPA. This indicates a strong positive association between physical activity and pain tolerance which was independent of time passing. Nevertheless, some findings indicated that LTPA might have a diminishing positive association over time, possibly due to ageing. As pain tolerance has been suggested to impact risk, or severity, of chronic pain, these results might suggest increasing PA levels as a possible non-pharmacological pathway towards reducing or preventing chronic pain.

## Supporting information

**S1 Table. Primary analysis sample missing data on baseline covariates (N = 6,864).** The Tromsø Study 2007–2016.
(DOCX)

**S2 Table. Secondary analysis sample missing data on baseline covariates (N = 10,732).** The Tromsø Study 2007–2016.
(DOCX)

**S3 Table. Regression coefficients with 95% confidence limits for the association between baseline levels of leisure-time physical activity and cold-pressor tolerance time (seconds) by sex or chronic pain[a].** The Tromsø Study 2007–2016. Mixed model Tobit regression with upper limit (censoring) = 106 s. for latent distribution of CPT outcome. Models were adjusted for measurement occasion, as well as baseline sex, age, and self-reported occupational PA level, education, alcohol consumption frequency, smoking status, health status, and chronic

pain. Significant results in bold.
(DOCX)

**S4 Table. Regression coefficients with 95% confidence limits for the association between baseline levels of leisure-time physical activity and cold-pressor tolerance time (seconds) according to sensitivity analyses.** The Tromsø Study 2007–2016. Censored estimates (all censored values included as is) by linear mixed models with random intercept. Models were adjusted for measurement occasion, as well as baseline sex, age, and self-reported occupational PA level, education, alcohol consumption frequency, smoking status, health status, and chronic pain. Significant results in bold.
(DOCX)

## Acknowledgments

We extend our most sincere gratitude to the staff and participants of the Tromsø Study for making this research possible.

## Author Contributions

**Conceptualization:** Anders Pedersen Årnes, Christopher Sievert Nielsen, Audun Stubhaug, Aslak Johansen, Bente Morseth, Tom Wilsgaard, Ólöf Anna Steingrímsdóttir.

**Data curation:** Anders Pedersen Årnes, Christopher Sievert Nielsen, Audun Stubhaug, Bente Morseth, Ólöf Anna Steingrímsdóttir.

**Formal analysis:** Anders Pedersen Årnes, Bjørn Heine Strand, Tom Wilsgaard, Ólöf Anna Steingrímsdóttir.

**Funding acquisition:** Anders Pedersen Årnes, Christopher Sievert Nielsen, Audun Stubhaug, Aslak Johansen, Ólöf Anna Steingrímsdóttir.

**Investigation:** Anders Pedersen Årnes, Mats Kirkeby Fjeld, Ólöf Anna Steingrímsdóttir.

**Methodology:** Anders Pedersen Årnes, Mats Kirkeby Fjeld, Bente Morseth, Bjørn Heine Strand, Tom Wilsgaard, Ólöf Anna Steingrímsdóttir.

**Project administration:** Anders Pedersen Årnes, Aslak Johansen, Ólöf Anna Steingrímsdóttir.

**Supervision:** Bente Morseth, Tom Wilsgaard, Ólöf Anna Steingrímsdóttir.

**Visualization:** Anders Pedersen Årnes.

**Writing – original draft:** Anders Pedersen Årnes.

**Writing – review & editing:** Anders Pedersen Årnes, Christopher Sievert Nielsen, Audun Stubhaug, Mats Kirkeby Fjeld, Aslak Johansen, Bente Morseth, Bjørn Heine Strand, Tom Wilsgaard, Ólöf Anna Steingrímsdóttir.

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
