## [Decision Letter · Decision Letter 0]

14 Feb 2023

PONE-D-23-01904Longitudinal relationships between habitual physical activity and pain tolerance in the general populationPLOS ONE

Dear Dr. Årnes,

Thank you for submitting your manuscript to PLOS ONE. After careful consideration, we feel that it has merit but does not fully meet PLOS ONE’s publication criteria as it currently stands. Therefore, we invite you to submit a revised version of the manuscript that addresses the points raised during the review process.

ACADEMIC EDITOR:Dear Authors, thanks for your submission. The paper was appreciated by the Reviewers. The final decision is a "Minor Revision". So, please, provide proper and accurate answers to the Reviewers's comments in order to improve you paper and then submit it again. 

We look forward to receiving your revised manuscript.

Kind regards,

Luca Russo, Ph.D.

Academic Editor

PLOS ONE

Reviewers' comments:

Reviewer's Responses to Questions

**Comments to the Author**

1. Is the manuscript technically sound, and do the data support the conclusions?

Reviewer #1: Yes

Reviewer #2: Yes

2. Has the statistical analysis been performed appropriately and rigorously? 

Reviewer #1: I Don't Know

Reviewer #2: Yes

3. Have the authors made all data underlying the findings in their manuscript fully available?

Reviewer #1: No

Reviewer #2: Yes

4. Is the manuscript presented in an intelligible fashion and written in standard English?

Reviewer #1: Yes

Reviewer #2: Yes

5. Review Comments to the Author

Reviewer #1: Dear Authors,

I think the paper is relevant, clear and well-written.

One question about the research-deign: would it be possible and relevant to add one more point to the two reported?

Moreover, I would suggest to present more synthetically and organically the discussion section to make it more readable.

Reviewer #2: The manuscript investigated the role of leisure-time physical activity (PA) in increasing the perceived chronic pain in a cohort of 10,732 individuals, of which 51% were women.The Authors concluded that being physically is associated with higher pain tolerance compared to being sedentary and that PA did not significantly moderate pain tolerance change over time as a possible effect of age.

First, I have to congratulate with the Authors for such an impressive study, whose results could have important socio-economic implications.

The introduction is concise and clear and the scope well presented.

Methods are adequately reported and the statistic approach correct. I just suggest the Authors to mover the Ethics paragraph just above the population description.

Results are clearly reposted and Discussion well supported by the main outcomes.

I have just to suggest the Authors a careful re-read of the manuscript for some possible typos.

6. PLOS authors have the option to publish the peer review history of their article (what does this mean?). If published, this will include your full peer review and any attached files.

Reviewer #1: No

Reviewer #2: No

---

## [Author Response · Author response to Decision Letter 0]

4 Mar 2023

Please find our current cover letter and full response to the editorial and review comments in the enclosed document "Response to reviewers".

---

## [Decision Letter · Decision Letter 1]

14 Apr 2023

Longitudinal relationships between habitual physical activity and pain tolerance in the general population

PONE-D-23-01904R1

Dear Dr. Årnes,

We’re pleased to inform you that your manuscript has been judged scientifically suitable for publication and will be formally accepted for publication once it meets all outstanding technical requirements.

Kind regards,

Luca Russo, Ph.D.

Academic Editor

PLOS ONE

Additional Editor Comments (optional):

I want to congratulate with the Authors. The Reviewers have positively evaluated the manuscript and the paper is accepted.

Reviewers' comments:

Reviewer's Responses to Questions

**Comments to the Author**

1. If the authors have adequately addressed your comments raised in a previous round of review and you feel that this manuscript is now acceptable for publication, you may indicate that here to bypass the “Comments to the Author” section, enter your conflict of interest statement in the “Confidential to Editor” section, and submit your "Accept" recommendation.

Reviewer #1: All comments have been addressed

Reviewer #2: All comments have been addressed

2. Is the manuscript technically sound, and do the data support the conclusions?

Reviewer #1: Yes

Reviewer #2: Yes

3. Has the statistical analysis been performed appropriately and rigorously? 

Reviewer #1: Yes

Reviewer #2: Yes

4. Have the authors made all data underlying the findings in their manuscript fully available?

Reviewer #1: Yes

Reviewer #2: Yes

5. Is the manuscript presented in an intelligible fashion and written in standard English?

Reviewer #1: Yes

Reviewer #2: Yes

6. Review Comments to the Author

Reviewer #1: Dear Authors,

Although I could not find the file with the answers to my specific comments, the changes you have made to the manuscript are in line with my suggestions.

Thank you

Best regards

Reviewer #2: I thank the Authors for considering all my previous suggestions. No further revisions are required.

7. PLOS authors have the option to publish the peer review history of their article (what does this mean?). If published, this will include your full peer review and any attached files.

Reviewer #1: **Yes: **Valerio Bonavolontà

Reviewer #2: No

---

## [Editor Report · Acceptance letter]

27 Apr 2023

PONE-D-23-01904R1 

Longitudinal relationships between habitual physical activity and pain tolerance in the general population 

Dear Dr. Årnes:

I'm pleased to inform you that your manuscript has been deemed suitable for publication in PLOS ONE. Congratulations! Your manuscript is now with our production department. 

Kind regards, 

on behalf of

Dr. Luca Russo 

Academic Editor

PLOS ONE